# Content Management Systems Performance and Compliance Assessment Based on a Data-Driven Search Engine Optimization Methodology

**Ioannis Drivas** \*,†,‡ , **Dimitrios Kouis** ‡ , **Daphne Kyriaki-Manessi** ‡ and **Georgios Giannakopoulos** ‡

Information Management Research Lab, Department of Archival, Library & Information Studies, University of West Attica, 12243 Athens, Greece; dkouis@uniwa.gr (D.K.); dkmanessi@uniwa.gr (D.K.-M.); gian@uniwa.gr (G.G.)

\* Correspondence: idrivas@uniwa.gr; Tel.: +30-210-538-5220
† Current address: 28, Agiou Spyridonos Str., 12243 Egaleo, Greece.
‡ These authors contributed equally to this work.

**Abstract:** While digitalization of cultural organizations is in full swing and growth, it is common knowledge that websites can be used as a beacon to expand the awareness and consideration of their services on the Web. Nevertheless, recent research results indicate the managerial difficulties in deploying strategies for expanding the discoverability, visibility, and accessibility of these websites. In this paper, a three-stage data-driven Search Engine Optimization schema is proposed to assess the performance of Libraries, Archives, and Museums websites (LAMs), thus helping administrators expand their discoverability, visibility, and accessibility within the Web realm. To do so, the authors examine the performance of 341 related websites from all over the world based on three different factors, Content Curation, Speed, and Security. In the first stage, a statistically reliable and consistent assessment schema for evaluating the SEO performance of LAMs websites through the integration of more than 30 variables is presented. Subsequently, the second stage involves a descriptive data summarization for initial performance estimations of the examined websites in each factor is taking place. In the third stage, predictive regression models are developed to understand and compare the SEO performance of three different Content Management Systems, namely the Drupal, WordPress, and custom approaches, that LAMs websites have adopted. The results of this study constitute a solid stepping-stone both for practitioners and researchers to adopt and improve such methods that focus on end-users and boost organizational structures and culture that relied on data-driven approaches for expanding the visibility of LAMs services.

**Keywords:** website seo; data driven seo; website performance; libraries; archives; museums; content management systems; cms comparison; drupal; wordpress

## 1. Introduction

It used to be that cultural heritage institutions, such as libraries, archives, and museums (also known as LAMs), were solely providing content and services within their physical realms. However, profound and rapid changes in societal, technological and economic context have shifted their traditional approach from keep and protectin physical sites to experience and engage within digital environments [1,2]. Web technologies, such as websites, constitute the main gateway for LAMs to provide access to content and collections among the interest parties. In this way, online users can interact with digital collections and cultivate cultural and educational background through online environments. Websites of such organizations help to drastically democratize high-quality information [3], expand capacities for visitation [4], and give the advantage for presenting cultural information that exceeds, by far, the available one at the physical sites [5].

Nevertheless, recent research efforts indicate difficulties in expanding these websites' discoverability, visibility, and accessibility. Several reasons are related to this situation,

both technical and behavioral. For example, the vast amount of information that these organizations possess lead to data overloading phenomena [2,6] and, hence, into difficulties in managing the voluminous cultural information. Furthermore, other works identify the administrators' limited managerial capabilities to understand the technical and behavioral web metrics and how they affect the discoverability and visibility levels of the websites [2,7]. Complexity gets higher, as only a few research efforts are trying to establish reliable and validated mechanisms that indicate factors that play an essential role in optimizing websites discoverability and visibility [8,9]. Lastly, the plurality of available Content Management Systems (CMSs) [10] to develop websites is another practical obstacle for expanding online awareness. Some of the CMSs present difficulties in administration and proper content curation, while the higher the content volume, the more complex its management becomes. In this respect, some of the cultural collections are not appropriately described in terms of their metadata. Thus, the probability of content that is not easily searchable and discoverable by users is high [11].

In this paper, we propose a data-driven Search Engine Optimization (SEO) methodology that is useful to assess with reliability and validity LAMs websites performance based on multiple related variables. To do so, we examine the performance of 341 libraries, archives, and museums websites from all over the world based on three different factors, Content Curation, Speed, and Security. One step further, predictive regression models are developed to understand and compare the SEO performance of two widely used CMSs, namely the Drupal, WordPress, and custom approaches (as a single category). To this end, this study is unfolded into four sections. In the next section, the related research efforts and the derived research gaps are entailed. Subsequently, in the third section, the proposed three-stage data-driven SEO methodology is presented. In the fourth section, the results of the proposed methodology are stated. More specifically, we present the statistical reliability and validity results of the proposed methodology, then the descriptive statistics for initial performance estimations per factor, and, lastly, the predictive regression models fit for each of the examined CMSs. In the last section, the discussion is taking place to highlight both the research and the study's practical contributions, thus expanding the knowledge and capabilities of interest parties or stakeholders.

## 2. Related Background

### 2.1. Libraries, Archives, and Museums Websites

Over the last few years, it is noted that multiple efforts have been made globally to digitize the cultural collections and artefacts and to enhance their visibility on the Web [12]. In this respect, several initiatives and research projects, such as Europeana [13], CrossCult [14], EMOTIVE [15], ViMM [16], Google Arts and Culture [17], etc., have emerged aiming to reinforce the online visibility of cultural institutions, focusing on improving awareness and consideration. One of the most important aspects of accessing digital cultural content through the Web is the institutional websites [2]. Over the years, research efforts have highlighted the advantages that websites offer to libraries, museums and archives as a gateway to communicate with their audiences [18]. But, if LAMs websites are not following proper content curation and technical compliance with search engines crawling mechanisms, then the collections' findability and visibility would be considerably low. In addition, LAMs websites that are well-structured provide educational and cultural information to users in a discoverable, usable and efficient manner [19].

As digital cultural content forms have evolved significantly in the last decade, websites offer the opportunity for rethinking and redesigning traditional services to diffuse novel cultural experiences for online visitors. In this sense, modern websites expand the visitation and audience navigation capabilities [20], allowing alternative types of interaction (compared to the physical attendance), with the collections, holdings, and artefacts [21]. This fact will enable LAMs to expand educational and cultural online content that sometimes exceeds the physical artefacts' volume. Online visitors often can seek information from a vast educational and cultural content database through the LAMs websites, satisfying their

needs. In addition, LAMs websites prepare users for their visitation to physical sites [22] and cultivate significant information gain in prior and post-visit experience level [23]. Especially for the prior and post-visit experience levels, findings of Pavlou [24] point out that the experience gained by the websites influences users' intentions to revisit them and subsequently to appreciate better the physical locations.

Although the research efforts mentioned above highlight the websites' role in expanding visibility and awareness of libraries, archives, and museums, the bounce rate is equal to 55%, indicating that one out of two users abandons the websites due to several issues, such as low content quality and quantity or poor usability level [6], even if the 62% of the online traffic to these organizations originates from search engines. In this respect, there is a need to develop a data-driven compliance assessment SEO methodology capable of evaluating LAMs websites to improve their findability and users' engagement with the content, hence decreasing the bounce rate. Therefore, a user-centric SEO should not emphasize predicting the position of a website in the results of a search engine [25] but improve the overall performance and usability for better user-content interaction. Immediate outcomes from such an approach are higher-ranking positions in search engines' results and more significant traffic from them [26,27].

The following sub-section unfolds research efforts that identify methods and strategies for improving LAMs websites visibility and findability based on a data-driven SEO methodology.

### 2.2. Prior Efforts and Research Gaps

Earlier research approaches and experiments utilized SEO strategies in developing a roadmap for improving content findability and visibility of LAMs. Within the sub-realm of academic libraries and SEO, Vállez and Ventura [28] analyzed the web visibility of 20 libraries. They discovered that their web sites suffer from low visibility, due to a strategic SEO schema absence. These findings are aligned with prior research indications that also highlight the need of library organizations' SEO policy [29–31]. In a similar manner, the study of Onaifo and Rasmussen [32] concluded that certain websites' characteristics, such as the proper indexing process, affect their ranking by search engines and should be considered within cultural institutions digital policies.

Furthermore, the Open SESMO project [33] suggested improving the visibility and search engines efficiency by indexing the linked and structured data stored in libraries repositories. For Alhuay-Quispe and colleagues [34], metadata quality and curation of collections is another feature that impacts drastically on content's visibility level. In an integrated approach of examining libraries, archives, and museums websites, Krstić and Masliković [2] indicated significant results. Based on their findings, archives portals have a very good level of SEO technical curation characteristics. Libraries lead in the occasional generation of new content development, while museums utilize social media platforms to expand their websites' volume and traffic quality. It was also highlighted the moderate or even low staff familiarization level with SEO strategies and web data analytics. Specifically, over 42% of the staff does not have access to analytics information, while 28% does not know if analytics are integrated into the web site or does not even know what analytics is.

Another drawback of the previous approaches for evaluating websites' performance was the limited amount of test cases. The main reason was technical constraints because LAMs public pages are usually part of higher level public organizations websites (e.g., city or county), making data collection challenging [32]. Subsequently, another reason is related to the voluminous content that these websites contain. Prior research efforts highlight the inversely proportional relationship between the content's size and the existence of strategic SEO schemes for optimizing findability and visibility [6]. The bigger the content volume, the higher the difficulties of performing SEO studies than involve multiple websites and examine technical data for analysis and interpretation.

Another common characteristic of the existing research efforts is the limited number of SEO variables that are examined for their impact on the findability and visibility of cultural institutions websites. Although there are many variables and technical factors that affect

website visibility and findability, administrators avoid using them in a data-driven SEO methodology due to the extra management complexity. In addition, administrators underestimate the importance of technical aspects and fail to define performance measurements and understand the cause-and-effect intercorrelations between those variables [7,35,36]. This kind of attitude could result in lower performance and content visibility of websites. Unless all the available technical data are not analyzed and interpreted sufficiently and not transformed into actionable insights for improving visibility and findability, the sustainability of the websites will remain low [37,38]. From another perspective, by selecting a small number of variables that influence websites performance, visibility could improve. Those SEO strategies could also be adopted quickly by the administrators [9,28]. However, in the era of big data, overall complexity increases within search engines algorithms the total number of variables that included affect the way of how websites are ranked [8,39]. In this sense, the higher the number of technical variables examined and curated, the more the possibilities to increase LAMs websites' findability by the search engines.

In terms of the variables, prior research efforts have not shown any SEO proposing high number of variables and consecutive results with statistical significance regarding reliability, cohesion and consistency. Such a reliable and consistent framework would allow replication and further experiments [40]. For example, it could be re-applied in websites of other domains, expecting relatively similar results and, thus, suggestions for overall optimization [41,42]. Practically, at a micro-level, this kind of approach could also work as a supportive evaluation tool to quantitatively measure each website's SEO performance and then to proceed with improvements both in content curation, structure, and usability [43].

A further investigation should be conducted regarding the adopted CMSs and their compliance with SEO factors. Prior examinations were relying heavily on challenges regarding security levels of CMSs [44], content copy prevention [45], better navigability [46], and recommending principles in specific case-studies for selecting a CMS [47]. Moreover, the research is limited in terms of CMSs comparisons for SEO purposes to expand websites' visibility and findability. Some exceptions are encountered at this specific topic [48,49]. Nevertheless, they have been conducted within a limited number of websites. Of course, the studies mentioned above avoid indicating "the best CMS" for deploying a website, as this outcome will have no practical use. Instead, they contribute to identifying technical errors in terms of content curation, speed performance, and security level that each adopted CMS assesses and subsequently impacts website visibility on the Web. There is no recent research approach that examines a large number of websites and how their adopted CMS performs under a plurality of multiple SEO variables to the best of our knowledge. Understanding how each website is developed based on the adopted CMS, the compliance of several technical variables, and how they impact the total SEO performance constitutes a more developer-centric approach that aims to improve user's experience and interaction. Since there is no best CMS, the aforementioned approach works as a practical toolbox for interested parties to understand the rectifying actions they need to perform in their websites.

Wrapping up, in the following table (Table 1), we provide the a summary of previous research focus and context and their perspective drawbacks and suggestions in relation to the present study. The latter is this present study's contribution on LAM's websites performance measurement context for greater visibility and findability on the Web. Then, the methodology used is presented.

**Table 1.** Reflections and context issues for SEO and new research avenues for the improvement of visibility and findability of websites on the Web.

| Research Context Issues | Contribution Needed |
|---|---|
| Staff low familiarization with SEO strategies and web analytics [2]. Reduced managerial capabilities in understanding SEO factors and the possible interrelationships between them [6,7,37,50]. | A novel SEO framework for better understanding and highlighting the importance of SEO for libraries, archives, and museums. |
| Deployed examinations regarding websites performance for SEO purposes with a limited number of cases, while also involving a small number of variables that affect visibility and findability [28,32–34]. | Further research efforts are needed to involve a large sample of websites in experiments for estimating SEO performance and involving additional variables that influence visibility and findability. |
| Lack of methodological framework to manage the voluminous size of websites and the impact on SEO for greater visibility and findability levels [6]. | A clear and understandable SEO framework is needed to manage websites with a large amount of content and a lack of proper technical curation. |
| Lack of SEO methodological framework that expresses statistical significance in terms of reliability, validity, and consistency within the involved variables for potential replication purposes [41,42]. | Establish an SEO data-driven framework and integration with other strategies for visibility and findability expansion on the Web. |
| Limited research efforts in CMSs comparisons and their impact on websites' overall SEO performance in terms of their content curation, speed loading time, and security level [48,49]. | Further examination of websites and the adopted CMSs is needed regarding their impact on SEO performance and identifying faults that impact the visibility and findability levels. |

## 3. Methodology

The purpose of the paper is to present a data-driven methodology that is capable of (a) providing a reliable assessment regarding LAMs websites SEO performance based on multiple variables that express validity, consistency and cohesion, (b) examining a large number of LAMs websites compared to prior research efforts with limited samples, and (c) making more understandable the performance of a website in terms of the adopted CMS. In this respect, we proceed into developing a three-stage methodological schema, as can be seen in Figure 1.

1.　In the first stage, we try to define a set of underlying variables for each of the proposed factors capable of affecting the total website SEO score. We describe three different factors that compile the Total Website SEO Performance construct, namely Content Curation, Speed, and Security factor. A further investigation and several reliability analysis tests are taking place regarding the internal consistency and the discriminant validity of the proposed variables and how they fit into each factor. This will expand potential research approaches to adopt this framework expecting relatively similar results concerning other domains' websites performance.

2.　In the second stage, a descriptive data summarization for each proposed factor takes place for initial performance estimations and exploratory purposes. Practically, this will allow administrators to understand their websites' SEO performance status through the extraction of detailed quantitative information for each of the involved variables set under the three different factors.

3.　At the third stage, assuming that there are differences among the adopted CMSs, we have developed diagnostic predictive models that estimate the possible potential impact of each factor on the SEO performance.

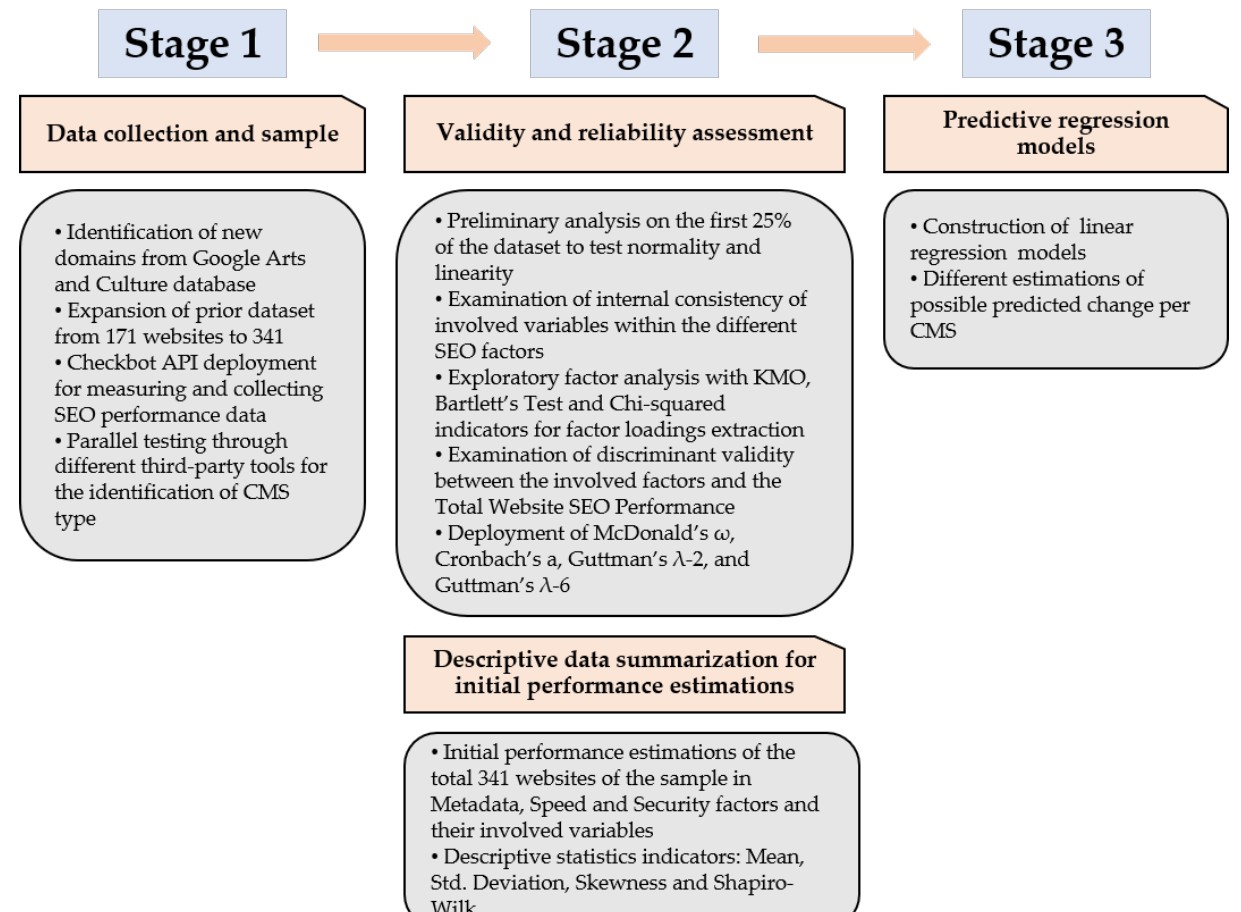

**Figure 1.** Proposed methodology schema to overcome the identified research barriers in expanding visibility and discoverability of LAMs websites.

In the next sub-section, the collection of data and the pre-processing are described extensively. In addition, we present the applied methods for testing the validity and reliability of the proposed factors.

### 3.1. Data Collection and Sample

Based on prior approach of the data gathering process [6], we expanded the range of the dataset while including even more organizations from the Google Arts and Culture Database [17]. Specifically, the number of the examined domains from LAMs websites was doubled (341), compared to the previous initial dataset (171). We used the Checkbot API to measure and collect websites compatibility on multiple SEO variables. Checkbot API indexes the website's code to find features capable of impacting SEO performance.

Several APIs extract technical websites performance estimations; however, we choose this one as it includes a greater plurality of variables than others in all the involved factors, that is, Content Curation, Speed, and Security. Each website has been tested using this tool; hence, 341 different tests were conducted. We settled up the tool to conduct each test at the maximum number of links allowed to be crawled, that is, 10,000 per test. In this way, we retrieved data about the overall websites' performance including their sub-pages, and not only the main domain names. A scale from 0 (lowest rate) to 100 (highest rate) was adopted for each examined variable. This constitutes a useful managerial indicator of dealing with the quantification of websites performance while avoiding complex measurement systems that are difficult to be adopted by administrators [36,50].

Note also that, when variables under consideration refer to lengths of characters, the optimal values are adopted from Google Search Engine Optimization (SEO) Starter Guide. For example, the Checkbot tool defines "Use optimal length titles" values as valid if they belong in the range of 10 to 60 characters [51]. In the same way, "Use optimal length H1 headings" should not exceed 70 characters, while the recommended page descriptions range is between 100 to 320 characters. To identify the type of CMS that the examined domains use, we performed a parallel cross-check using three different tools, namely WhatCMS, CMSDetector, and CMSDetect.

After collecting 341 LAMs websites and their performance in multiple variables, further data pre-processing took place. The majority of the examined websites (264 out of 341) used Drupal, WordPress or a custom approach as their CMS. The rest of the domains used other CMS types, such as Adobe Dreamweaver, Ruby on Rails, or Joomla. Nevertheless, it was not practically possible to analyze the less frequent appearing CMSs types due to the low sample size.

### 3.2. Validity and Reliability Assessment

After the collection, the pre-processing, and the dataset's organization, statistical analysis was performed to validate its reliability and consistency. A preliminary analysis was conducted on the first 25% of the dataset to ensure the assumption that the gathered sample express normality and linearity in its nature [52]. To test the normality and linearity of the dataset, the Shapiro-Wilk test and the p values were calculated as the most powerful indicators compared to others tests [53]. In addition, we involved skewness measurement in understanding the tendency of each variable along the percentage scale (0–100). For example, negative skewness for the involved variables indicates that their values tend to 100. In the opposite direction, positive skewness points out that their values tend to 0 [54]. Based on the skewness tendency in each of the SEO variables, a more precise estimation is achieved concerning the total performance of the examined websites.

To categorize the extracted variables of the dataset into specific valid and reliable factors (Content Curation, Speed, and Security), an initial exploratory factor analysis (EFA) was performed. Under the EFA process, the Kaiser-Meyer-Olkin (KMO), the Bartlett's test of Sphericity and the Chi-Squared were performed as indicators for testing the fit of the involved variables into each one of the proposed factors [55]. Certain variables with loadings below the limit of 0.500 were excluded from each of the proposed factors. These variables are represented with a strike-through line in Table 2.

One step further, to testify the internal consistency and the discriminant validity of the involved variables within each factor, we implemented four different verification tests, namely the McDonald's $\omega$, Cronbach's a, and Guttman's $\lambda$-2 and $\lambda$-6 accordingly. McDonald's $\omega$ estimates the strength of association between the involved variables, meaning the closer the value to 1 the greater the association strength between the variables, and vice-versa [56]. Cronbach's a test was used to assess the acceptance level of each of the proposed factors [57]. Guttman's $\lambda$-2 works supportively to Cronbach's a test estimating the variance trustworthiness among the collected variables of the percentage scale [58]. We also used Guttman's $\lambda$-6 test to examine if each of the individual website performance variables values differ significantly and if the percentage values from 0 to 100 correspond to true scores. The use of Guttman's $\lambda$-6 test was also necessary as one of the main goals of this research effort was to develop linear predictive models for total Website SEO performance in different CMSs. This indicator calculates the variance in each variable that could be involved in regression models [59].

**Table 2.** Exploratory factorial analysis results and loadings per variable.

| Content Curation Factor | | Speed Factor | | Security Factor | |
|---|---|---|---|---|---|
| Variables | Variable Loading | Variables | Variable Loading | Variables | Variable Loading |
| Set page titles | ~~0.499~~ | Avoid temporary redirects | 0.565 | Use content sniffing protection | 0.594 |
| Use optimal length titles | ~~0.311~~ | Use compression | 0.592 | Use clickjack protection | 0.590 |
| Use unique titles | 0.661 | Use minification | 0.584 | Use HTTPS | 0.680 |
| Set H1 headings | 0.630 | Avoid render-blocking JavaScript | 0.545 | Hide server version data | 0.611 |
| Use one H1 heading per page | 0.656 | Use long caching times | 0.596 | Avoid mixed content | 0.756 |
| Use optimal length H1 headings | 0.627 | Avoid duplicate resources | 0.605 | Use HSTS | 0.821 |
| Use unique H1 headings | 0.706 | Avoid plugins | 0.528 | Use HSTS preload | ~~0.487~~ |
| Set page descriptions | 0.653 | Avoid resource redirects | 0.571 | Use XSS protection | 0.761 |
| Use optimal length descriptions | 0.674 | Use valid HTML | 0.569 | Use secure password forms | 0.504 |
| Use unique descriptions | 0.823 | Use valid CSS | ~~0.487~~ | Set MIME types | ~~0.437~~ |
| Set canonical URLs | 0.760 | Avoid recompressing data | ~~0.465~~ | | |
| Avoid duplicate page content | 0.813 | Avoid excessive inline JavaScript | 0.558 | | |
| Avoid thin content pages | 0.791 | Avoid excessive inline CSS | 0.523 | | |
| Set image ALT text | 0.753 | Avoid CSS @import | 0.529 | | |
| Set mobile scaling | 0.806 | Avoid internal link redirects | ~~0.477~~ | | |
| Use short URLs | 0.770 | | | | |
| 0.792 *<0.001 ** <0.001 *** | | 0.582 * <0.001 ** <0.023 *** | | 0.627 * <0.001 ** <0.001 *** | |

* KMO, ** Bartlett's Sphericity *p*-value, *** Chi-squared *p*-value.

### 3.3. Predictive Regression Models

We performed linear regression models to estimate the potential change of the Total Website SEO Performance if the three factors increase by one percentage unit. Separate linear regressions were performed for every type of the examined CMSs to assess the Total Website SEO Performance impact comparatively. For example, different values of Total Website SEO Performance changes were observed, per CMS type, when a factor increase was applied (e.g., Content Curation by a percentage unit). The practical contribution of these tests was to assist in the development of a reliable, data-driven SEO compliance assessment methodology. Hence, the proposed measurement system's outcome would be capable of being adopted and applied by practitioners in LAMs, enhancing the visibility and findability of their online content.

In the next section, the results of the proposed methodology experiments are presented.

### 4. Results

### 4.1. Validation of the Proposed Factors

Based on the first step of the proposed methodology, reliability tests took place to validate the internal consistency and cohesion of the proposed factors. At the initial stage, an Exploratory Factor Analysis (EFA) was performed to understand the overall validity of

the proposed factors and if their variables are suitable to be categorized. Table 2 indicates the EFA results and the loadings per variable for each factor. As mentioned earlier in the methodology chapter, variable loadings below the limit of 0.500 were excluded from the proposed model. KMO, Bartlett's Sphericity, and Chi-squared tests extracted sufficient values indicating that the proposed variables could be categorized into three different factors, the Content Curation, the Speed, and the Security.

Some variables seem to influence the consistency and cohesion of each factor at a higher level rather than others. To refer to some of them, within Content Curation factor, the variables of *use unique descriptions*, *avoid duplicate page content*, *set mobile scaling*, *avoid thin content pages*, and *use short URLs* extracted loadings greater than 0.750. On the other hand, the variables *set page titles* and *use optimal length titles* were excluded due to low factor loadings. At an overall level, the Content Curation factor articulates a KMO value at 0.792, while both Bartlett's test and Chi-squared *p*-values extracted very high statistical significance at <0.001. Subsequently, compared to the Content Curation factor, the Speed factor indicates lower sampling adequacy and variables loadings. For instance, KMO value is 0.582, Bartlett's test *p*-value is <0.001 and Chi-squared *p*-value is <0.023. Some variables seem to receive higher influence by the factor itself with loadings greater than 0.580, such as the *use compression*, *use minification*, *use long caching times*, and *avoid duplicate resources*. In the third column, the variables loadings of Security factor are presented, while extracting KMO value is 0.627 and *p*-values for both Bartlett's and Chi-squared tests are <0.001. The variables *use HSTS*, *avoid mixed content*, and *use XSS protection* extracted higher influence by this factor, while *use HSTS preload* and *set MIME types* excluded due to low values of loadings.

Table 3 depicts the internal consistency and the discriminant validity of the proposed model that aims to quantify through a percentage way the total SEO performance of websites.

**Table 3.** Internal consistency and discriminant validity of the proposed Total Website SEO Performance.

| Factors | McDonald's $\omega$ | Cronbach's $\alpha$ | Guttman's $\lambda$-2 | Guttman's $\lambda$-6 |
|---|---|---|---|---|
| Content Curation | 0.762 | 0.765 | 0.781 | 0.824 |
| Speed | 0.452 | 0.454 | 0.480 | 0.520 |
| Security | 0.613 | 0.604 | 0.628 | 0.605 |

As it can be seen in Table 3, Content Curation factor and its involved variables designate high reliability and internal consistency values in all different measurements ranging from 0.762 (McDonald's $\omega$) to 0.824 (McDonald's $\omega$). Speed factor and the involved variables extracted lower reliability and internal consistency values ranging from 0.452 (McDonald's $\omega$) to 0.520 (Guttman's $\lambda$-6). Lastly, the Security factor articulates sufficient reliability and internal consistency values ranging from 0.604 (Cronbach's a) up to 0.628 (Guttman's $\lambda$-2). Taking into consideration statistical findings [40–42] and the Total Website SEO Performance and its involved factors tests' values presented above, there is a high probability that similar reliability and internal consistency values would also apply in websites of different domains.

### 4.2. Descriptive Data Summarization for Initial Performance Estimations

In this sub-section, we present the descriptive results of the examined 341 websites and how they perform among the different variables in each factor. This stage works as a bootstrap for initial exploratory purposes regarding the performance of the examined websites based on the variables contained in each factor (Content Curation factor: 14 variables, Speed factor: 12 variables, and Security factor: 8 variables). In the upcoming tables (Tables 4–6), descriptive statistics are presented. Statistical measurements of mean, standard deviation, skewness, and Shapiro-Wilk are calculated.

Furthermore, in Figure 2, we present the different CMS types that the 341 websites used. Within Figure 2, the horizontal axis articulates the name of the CMSs. We identified 35 different CMS types. The number in each bar indicates the number of websites that

selected a specific CMS type. For example, 104 websites use a custom website development approach, 88 Drupal, 72 WordPress, etc. Lastly, for space-save reasons, in the last bar at the right, we include 24 websites out of 341 that use other CMS types.

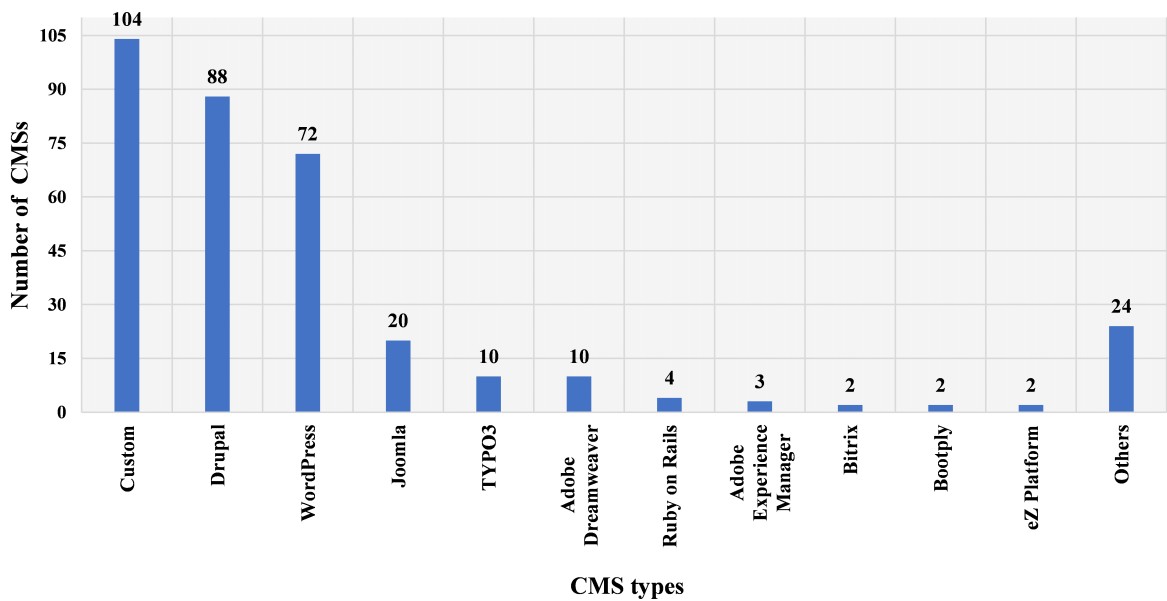

**Figure 2.** Bar chart included the number of CMSs types among the 341 examined websites.

**Table 4.** Descriptive statistics results for the variables included within Content Curation factor.

| Variables | Mean | Std. Deviation | Skewness | Shapiro-Wilk |
|---|---|---|---|---|
| Use unique titles | 57.164 | 33.4 | −0.389 | 0.897 |
| Set H1 headings | 72.258 | 35.934 | −1.135 | 0.720 |
| Use one H1 heading per page | 58.012 | 39.847 | −0.425 | 0.814 |
| Use optimal length H1 headings | 70.853 | 34.742 | −1.12 | 0.763 |
| Use unique H1 headings | 34.982 | 34.936 | 0.415 | 0.838 |
| Set page descriptions | 36.396 | 40.23 | 0.523 | 0.78 |
| Use optimal length descriptions | 19.537 | 29.302 | 1.494 | 0.709 |
| Use unique descriptions | 17.947 | 28.711 | 1.532 | 0.678 |
| Set canonical URLs | 26.07 | 41.107 | 1.047 | 0.613 |
| Avoid duplicate page content | 76.962 | 27.661 | −1.339 | 0.79 |
| Avoid thin content pages | 68.164 | 34.975 | −0.807 | 0.813 |
| Set image ALT text | 67.513 | 40.601 | −0.846 | 0.718 |
| Set mobile scaling | 82.167 | 35.449 | −1.771 | 0.527 |
| Use short URLs | 78.337 | 26.985 | −1.392 | 0.77 |

In Table 4, the descriptive statistics about the Content Curation factor and the involved variables are presented. All the variables articulate sufficient Shapiro-Wilk values ranging from 0.527 (*Set mobile scaling*) up to 0.897 (*use unique titles*), indicating, in this way, normality to their distribution. Moreover, 5 out of 13 variables extracted positive skewness values showing percentages close to 0. For instance, the *use unique descriptions* variable presents the lowest percentage mean value (17.947/100) and the highest positive skewness (1.532). Within the same line, variables with low percentage mean values are the *use optimal length descriptions* (19.537/100) and the *set canonical URLs* (26.07/100). On the other hand, the examined websites showed significant positive indications regarding their mobile responsiveness curation of their content, exhibiting a mean value of *set mobile scaling* variable at 82.167/100 and the highest negative skewness value (−1.771). Some other

variables present very high mean values closer to 100 of the percentage scale, namely, the *avoid duplicate page content* (76.962/100) and the *use short URLs* (78.337/100).

**Table 5.** Descriptive statistics results for the variables included within Speed factor.

| Variables | Mean | Std. Deviation | Skewness | Shapiro-Wilk |
|---|---|---|---|---|
| Avoid_temporary redirects | 71.012 | 37.511 | −0.923 | 0.742 |
| Use compression | 73.93 | 41.85 | −1.126 | 0.593 |
| Use minification | 55.466 | 25.025 | 0.036 | 0.979 |
| Avoid render-blocking JavaScript | 16.792 | 34.533 | 1.804 | 0.516 |
| Use long caching times | 41.079 | 44.76 | 0.346 | 0.721 |
| Avoid duplicate resources | 80.754 | 31.58 | −1.499 | 0.652 |
| Avoid plugins | 98.689 | 8.848 | −9.889 | 0.125 |
| Avoid resource redirects | 82.713 | 34.193 | −1.806 | 0.533 |
| Use valid HTML | 30.789 | 39.613 | 0.726 | 0.721 |
| Avoid excessive inline JavaScript | 81.683 | 34.298 | −1.709 | 0.566 |
| Avoid excessive inline CSS | 97.123 | 15.807 | −5.887 | 0.169 |
| Avoid CSS @import | 93.006 | 17.327 | −3.827 | 0.519 |

Table 5 contains the descriptive results of the Speed factor. It is noted that some variables resulted lower mean percentage values below 50, such as the *use valid HTML* (30.789), the *use long catching times* (41.079), and the *avoid render blocking JavaScript* (16.792). Furthermore, regarding data normality, some variables indicated low Shapiro-Wilk values as the *avoid plugins* (0.125) and the *avoid excessive inline CSS* (0.169). According to prior investigations, data normality constitutes one of the most vital prerequisites for developing prediction models with sufficient statistical validity [60]. However, more recent approaches indicated that, in large-sample sizes, violations in data normality within some individual variables do not noticeably impact results [61]. Since the examined data conform with the large sample size principles for developing prediction models [62,63], we chose not to exclude these two variables from their involvement in the prediction models that follow. Moreover, to test our selection, regression experiments were repeated both with and without the two variables with zero or statistically non-significant differences.

**Table 6.** Descriptive statistics results for the variables included within Security factor.

| Variables | Mean | Std. Deviation | Skewness | Shapiro-Wilk |
|---|---|---|---|---|
| Use content sniffing protection | 27.44 | 39.151 | 0.967 | 0.684 |
| Use clickjack protection | 40.305 | 47.868 | 0.383 | 0.653 |
| Use HTTPS | 76.05 | 41.919 | −1.232 | 0.549 |
| Hide server version data | 56.44 | 45.489 | −0.231 | 0.727 |
| Avoid mixed content | 90.19 | 20.775 | −2.704 | 0.537 |
| Use HSTS | 18.692 | 36.277 | 1.597 | 0.541 |
| Use XSS protection | 23.378 | 38.13 | 1.178 | 0.621 |
| Use secure password forms | 85.088 | 30.547 | −1.965 | 0.542 |

Regarding the Security factor (Table 6), it is noted that most of the websites *use HTTPS* protocol (76.05/100). In the same line, *avoid mixed content* and *use secure password forms* have high mean values (90.19/100 and 85.088/100), indicating that administrators' willingness to curate security issues properly within the LAMs websites cases. Controversially, some other variables within this factor designate lower percentage values, such as the *use HSTS*, the *use XSS protection*, and the *use content sniffing protection*.

### 4.3. Predictive Regression Models

The results of the predictive regression models are presenting starting with the most often used CMS type (in our case, custom CMSs). For all CMSs types the estimated

coefficients resulted in minimum to zero probability of being non-reliable. Additionally, all the values indicated high statistical significance with $p < 0.001$. Supportively to the $p$-value, we also included F value to prove that the proposed predictive models can reject the null hypothesis. That is, all the regression coefficients are equal to zero, or, in other words, the proposed models do not have predictive discriminant capability. The higher the F value, the better the predictive capability [64].

In Table 7, the results of the regression equation regarding the Custom CMS and the predicted change in constant value of Total Website SEO Performance is presented.

**Table 7.** Regression equation output of websites with Custom CMS and their change in each proposed factor.

| Variable | Coefficient | $R^2$ | F | *p*-Value |
|---|---|---|---|---|
| Constant (Total Website SEO Performance) | 20.953 | 0.786 | 414.469 | <0.001 |
| *Content Curation Factor* | 0.647 | | | |
| Constant | 25.703 | 0.279 | 43.783 | <0.001 |
| *Speed Factor* | 0.481 | | | |
| Constant | 44.109 | 0.286 | 45.216 | <0.001 |
| *Security Factor* | 0.317 | | | |

More specifically, based on the predictive regression model, Total Website SEO Performance for custom CMSs is affected by alterations of Content Curation, Speed, and Security factors values change. A significant regression equation was observed with $p \leq 0.001$ and $R^2$ of 0.786 between the Content Curation factor and the Total Website SEO score. In more detail, the Total Website SEO Performance (constant) rises by 0.647 for every percentage point increment of the Content Curation factor. Regarding the Speed factor case ($p < 0.001$ and $R^2$ of 0.279), results showed that the Total Website SEO Performance increases by up to 0.481 for every percentage point addition. Lastly, the Security factor ($p < 0.001$ and $R^2$ of 0.286) constitutes another important dimension capable of changing the percentage value of the Total Website SEO Performance score by up to 0.309 for every percentage point gain.

In the next table (Table 8), the regression model fit is presented regarding the predicted change of Total Website SEO Performance in Drupal type of CMSs.

**Table 8.** Regression equation output of websites with Drupal CMS and their change in each proposed factor.

| Variable | Coefficient | $R^2$ | F | *p*-Value |
|---|---|---|---|---|
| Constant (Total Website SEO Performance) | 23.097 | 0.688 | 213.727 | <0.001 |
| *Content Curation Factor* | 0.664 | | | |
| Constant | 45.357 | 0.155 | 17.818 | <0.001 |
| *Speed Factor* | 0.309 | | | |
| Constant | 51.035 | 0.329 | 47.481 | <0.001 |
| *Security Factor* | 0.270 | | | |

In almost the same line with the custom CMSs, Drupal websites also present a significant regression equation with $p \leq 0.001$ and $R^2$ of 0.688 between the Content Curation factor and the Total Website SEO score. Specifically, the Total Website SEO Performance (constant) rises by 0.664 for every percentage point increment of the Content Curation factor. Regarding the Speed factor case ($p < 0.001$ and $R^2$ of 0.155), results showed that the Total Website SEO Performance increases by up to 0.309. Lastly, when Security factor ($p < 0.001$ and $R^2$ of 0.329) increased by one percentage point, the Total Website SEO Performance score rises by 0.270.

In the last table (Table 9), the regression equation model fit is presented regarding the WordPress types of CMSs. Compared to the custom and Drupal CMS types, WordPress has

the highest increment (0.759) of Total Website SEO Performance when the Content Curation factor ($p < 0.001$ and $R^2$ of 0.155) is increased by one percentage point. The increment step values for the Speed and Security factor are 0.491 and 0.295 accordingly.

**Table 9.** Regression equation output of websites with WordPress CMS and their change in each proposed factor.

| Variable | Coefficient | $R^2$ | F | *p*-Value |
|---|---|---|---|---|
| Constant (Total Website SEO Performance) | 13.369 | 0.759 | 220.858 | <0.001 |
| *Content Curation Factor* | 0.743 | | | |
| Constant | 33.887 | 0.399 | 46.396 | <0.001 |
| *Speed Factor* | 0.491 | | | |
| Constant | 50.542 | 0.295 | 24.935 | <0.001 |
| *Security Factor* | 0.295 | | | |

From an overall perspective, there are some important differences between custom, Drupal and WordPress CMSs regarding the three different factors and the predicted constants (see Figure 3). WordPress CMS has the highest predicted change regarding the Content Curation factor with a value of 0.743, followed by Drupal (0.664) and custom CMSs (0.647). This practically means that WordPress administrators are in a more favorable position regarding their efforts on Content Curation variables to improve the Total Website SEO Performance rather than Drupal or custom CMSs. WordPress CMSs (0.491) barely outperformed custom approaches (0.481) in terms of the Speed factor. At the same time, Drupal received the lowest predicted value of change at 0.309. Lastly, in terms of the Security factor, custom CMSs approaches received the highest predicted value of change, reaching up to 0.317, followed by WordPress (0.295) and Drupal (0.27).

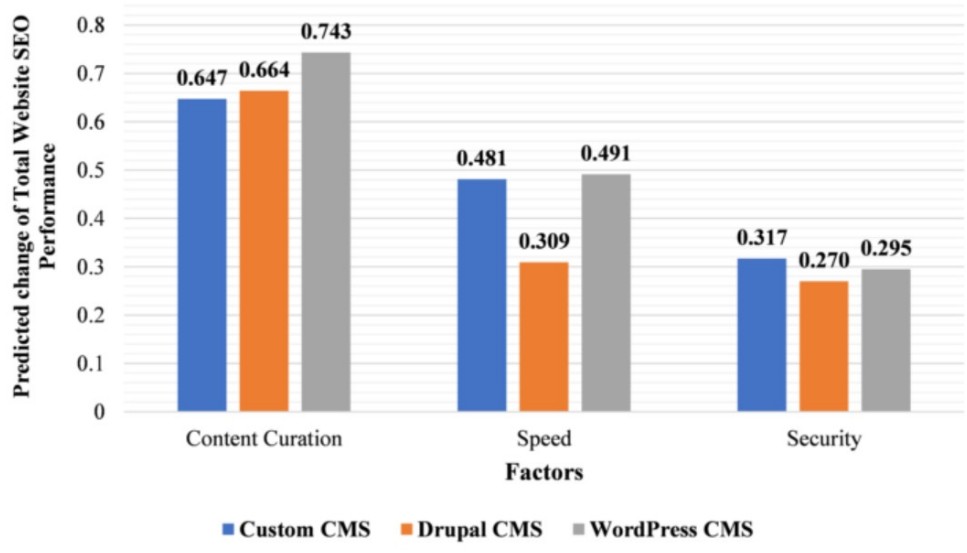

**Figure 3.** Comparison among the factors and their potential predicted change to Total Website SEO Performance.

Finally, in terms of the $R^2$, we have already noticed that some factors in specific CMSs indicated lower values compared to the other cases (e.g., $R^2$ value of 0.155 in Speed factor at Table 8). This is not a non-expected result as, the more the variables, the higher the model fit and, hence, the $R^2$ values [65]. Subsequently, prior investigations articulate that search engines algorithms compute a massive amount of variables within the SEO context [8,66]. In this respect, apart from the variables involved in each factor, searching for other complementary related variables to enhance the internal consistency of the proposed model both for Speed [67] and Security [68] is a future research goal.

## 5. Discussion

Published statistics indicate that users' needs for accessing cultural heritage content online have increased significantly in the last three years over the EU [69]. Nevertheless, recent reports also indicated the low percentages of expenditures in cultural heritage, including web presence activities [70]. In this sense, low-cost and easy to implement solutions for LAMs administrators to redefine a new era of coming up with innovative forms of digital heritage content delivered through the web is imperative. Prior research efforts also report the need to expand capacities for visitation [4] and increase the democratization and dissemination of cultural information to stakeholders and interest parties [3]. Several studies, including ours, proposed the user-centric data-driven SEO methods as a cost-efficient marketing approach to optimize LAMs websites toward expanding their discoverability and visibility of their collections and holdings [2,6,28,34]. Besides, SEO efforts constitute an essential supplementary step to cultivating sustainable marketing endeavors to promote cultural collections and holdings. More specifically, if SEO efforts do not accompany promotional activities, websites will present a low level of usability and content curation resulting in high bounce rates. In contrast, if administrators pay greater attention to SEO factors, the visibility and findability will be improved, and end-users will have a better prior and post-visit experience regarding cultural content [23,24]. In this sense, this present study focused firstly at providing a reliable and consistent assessment schema for evaluating LAMs websites SEO performance by integrating multiple variables grouped into three factors, namely Content Curation, Speed, and Security. Secondly, we tested the proposed schema regarding SEO performance assessment at a large set of LAMs websites (341 in total). Thirdly, by relying on predictive linear models, we calculated the possible changes in Content Curation, Speed, and Security factors to the Total Website SEO Performance per different CMSs' type. To the best of our knowledge, there is no significant prior research approach that embraces both SEO variables' reliability, the involvement of a large number of them, and estimations of predicted change per CMS. Therefore, this framework can cover prior gaps regarding the reliability of web metrics used during SEO performance and compliance assessment.

Our method to extract SEO performance metrics could be deployed on websites that adopt different CMSs, both in micro and marco-level. More specifically, by utilizing the second stage of the proposed methodology, the descriptive data summarizations of the involved variables for initial performance estimations could be calculated. This gives the advantage to validate and improve certain cultural content and parts of a website and their SEO performance (micro-level), and the website as a whole, its content and capabilities, while eliminating possible technical vulnerabilities (macro-level). In addition, based on a waterfall perspective, this strengthens further the efforts to assess and manage more efficiently LAMs websites due to their large-scale and voluminous cultural content [5,6].

In terms of the CMSs comparison, we expand the knowledge of the related community not only to compare CMSs regarding specific aspects, such as security [44], illegal content copy [45], and navigability [46], but also for SEO performance issues. There are two CMS-centric SEO research approaches in the past [48,49], however, with limited websites cases for comparative purposes and a relatively limited number of extracted valuable insights for administrators. In our study, we defined through regression equation models the possible predicted change for the three factors that impact the CMSs Total SEO Performance. The experiments for our sample indicated that WordPress CMS has a clear advantage in Content Curation factor optimization. Furthermore, WordPress slightly outperforms custom CMSs in terms of the Speed factor, while custom approaches topped WordPress and Drupal as to Security.

Taking into consideration prior approaches, there is some evidence that LAMs staff express a low level of familiarization with SEO strategies and web analytics [2,7,26,37,50]. This paper highlighted the importance of the three factors and their variables for developing a user-centric, data-driven SEO strategy that could expand the knowledge and familiarization of LAMs administrators. For example, we included a plurality of variables

that administrators can focus on to increase the usability of LAMs websites. From a technical and administrative perspective, our methodology could help developers improve the CMSs platforms capabilities by quickly discovering and rectifying SEO features related to Content Curation, Speed, and Security, within the deployed CMS platform. Based on this, some CMSs offer an easier way to improve specific factors than others [71]. For instance, the WordPress platform provides a more straightforward layout to improve Content Curation variables, resulting in a higher score to this factor compared to other CMSs. Therefore, developers could utilize the results of this study to optimize the existing CMSs usability and their compliance with the factors improving this way the Total Websites SEO performance.

*Future Implications*

Constructing a statistically significant model and its validation through the related indicators forms an essential step for similar potential investigations. It is in our intentions to apply our novel research approach to other web business sectors (e.g., e-shops, news etc.), expecting similar reliability and consistency significance results [41,42]. At the same time, our framework offers the opportunity to estimate initial SEO performance through the stage of descriptive data summarization. Following this assertion, we also encourage prior approaches within the realm of LAMs to re-examine discoverability and visibility levels of related websites based on our data-driven SEO framework [2,28,29,32,34].

Lastly, we have already started to build a quantitative methodological tool to examine the technology acceptance levels of the proposed model focusing on LAMs websites administrators. This will help develop a holistic user-centric data-driven SEO methodology involving both websites performance data and LAMs administrators opinion.

**Author Contributions:** All authors have contributed equally. All authors have read and agreed to the published version of the manuscript.

**Funding:** The APC was funded by the Information Management Research Lab, Department of Archival, Library and Information Studies, University of West Attica.

**Institutional Review Board Statement:** Not applicable.

**Informed Consent Statement:** Not applicable.

**Data Availability Statement:** Data available in a publicly accessible repository. The data presented in this study are openly available in Zenodo at: https://doi.org/10.5281/zenodo.4992230.

**Conflicts of Interest:** The authors declare no conflict of interest.

**Abbreviations**

The following abbreviations are used in this manuscript:

| | |
|---|---|
| LAMs | Libraries, Archives and Museums |
| SEO | Search Engine Optimization |
| CMS | Content Management Systems |
| EMOTIVE | Emotive Virtual cultural Experiences through personalized storytelling |
| ViMM | Virtual Multimodal Museum |
| KMO | Kaiser-Meyer-Olkin |
| EFA | Exploratory Factor Analysis |
| HTTPS | Hypertext Transfer Protocol Secure |
| HTML | HyperText Markup Language |
| HSTS | HTTP Strict Transport Security |
| XSS | Cross-site scripting |
| MIME | Multipurpose Internet Mail Extensions |
| URL | Uniform Resource Locator |
| CSS | Cascading Style Sheets |
| ALT | Alternative Text |

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
