# Peer review of "Content Management Systems Performance and Compliance Assessment Based on a Data-Driven Search Engine Optimization Methodology"

_information, doi:10.3390/info12070259_

Round 1

Reviewer 1 Report

The subject of the paper “Content Management Systems Performance and Compliance Assessment based on a Data-Driven Search Engine Optimization Methodology” is timely and valuable to the audience of the Information. Researchers presented results from testing 341 Library, Museum, and Archives websites in terms of content, speed and security.

Overall, the paper is well structured, reads quite well, and covers the existing literature quite well. The analysis of the data is interesting and well documented. However, in my view, some major amendments are required prior to publication.

The subset Metadata is actually not metadata. To naming it in this way is confusing since here are most elements that are clearly visible on the website. Not only in the meta of the website code. What is the optimal length description? How do you set the optimum?

The same applies to subset Speed. Speed factor consists of three major groups of other key factors influencing speed: server-side, download/transmission time, and browser render time. Here, most of your factors belong to the „download/transmission” group.

Compare website load time factors in „Content Delivery Networks in Cross-border e-Commerce”. IJACSA, 11(4), 19-27. https://doi.org/10.14569/IJACSA.2020.0110404

Using Checkbot API, have you rendered a website? Was it included in the speed area? Did you check only main URLs like the main domain name, or you also tested subpages?

In Table 4, what is the value? The mean length of characters? The mean number of occurrences?

In line 143 sentence is not finished.

The manuscript is missing at least a list of tested websites as supplementary material.

Author Response

Letter to the reviewers

First of all, we would like to express our appreciation to the reviewers for their valuable comments and observations. Our answers for all reviewers’ comments follow:

Reviewer 1

Reviewer 1 Comment 1:

The subject of the paper “Content Management Systems Performance and Compliance Assessment based on a Data-Driven Search Engine Optimization Methodology” is timely and valuable to the audience of the Information. Researchers presented results from testing 341 Library, Museum, and Archives websites in terms of content, speed and security.

Overall, the paper is well structured, reads quite well, and covers the existing literature quite well. The analysis of the data is interesting and well documented. However, in my view, some major amendments are required prior to publication.

The subset Metadata is actually not metadata. To naming it in this way is confusing since here are most elements that are clearly visible on the website. Not only in the meta of the website code. What is the optimal length description? How do you set the optimum?

Response:

We accepted the reviewer's comment and replaced the Metadata term with Content Curation as it provides a more accurate description of the variables included within this factor.

For variables that refer to lengths of characters, the text of the paper was updated accordingly to answer the reviewer comments (Section 3.1).

“Note also that when variables under consideration refer to lengths of characters, the optimal values are adopted from Google Search Engine Optimization (SEO) Starter Guide. For example, the Checkbot tool defines "Use optimal length titles" values as valid if they belong in the range of 10 to 60 characters [51]. In the same way, "Use optimal length H1 headings" should not exceed 70 characters, while the recommended page descriptions range is between 100 to 320 characters.”

Reviewer 1 Comment 2:

The same applies to subset Speed. Speed factor consists of three major groups of other key factors influencing speed: server-side, download/transmission time, and browser render time. Here, most of your factors belong to the „download/transmission” group. Compare website load time factors in „Content Delivery Networks in Cross-border e-Commerce”. IJACSA, 11(4), 19-27. https://doi.org/10.14569/IJACSA.2020.0110404

Response:

We accept the reviewer's comment, and we have updated the text of the paper accordingly by taking into account the proposed reference (at the end of Section 4.3).

“In this respect, apart from the variables involved in each factor, searching for other complementary related variables to enhance the internal consistency of the proposed model both for Speed [67] and Security [68] is a future research goal.”

Reviewer 1 Comment 3:

Using Checkbot API, have you rendered a website? Was it included in the speed area? Did you check only main URLs like the main domain name, or you also tested subpages?

Response:

Thank you for your valuable remark here. We used the Checkbot API for retrieving data also for speed and security performance issues. We have updated the text of the paper accordingly to provide further information to cover this gap (Section 3.1).

“Several APIs extract technical websites performance estimations; however, we choose this one as it includes a greater plurality of variables than others in all the involved factors, that is, Content Curation, Speed and Security. Each website has been tested using this tool, hence, 341 different tests were conducted. We settled up the tool to conduct each test at the maximum number of links allowed to be crawled, that is 10.000 per test. In this way, we retrieved data about the overall websites’ performance including their sub-pages, and not only the main domain names.“

Reviewer 1 Comment 4:

In Table 4, what is the value? The mean length of characters? The mean number of occurrences?

Response: See response to Reviewer 1 comment 1

Reviewer 1 Comment 5:

In line 143 sentence is not finished.

Response: This sentence has been deleted.

Reviewer 1 Comment 6:

The manuscript is missing at least a list of tested websites as supplementary material.

Response: We have uploaded the full dataset to Zenodo Repository (http://doi.org/10.5281/zenodo.4992230). According to the Journal guidelines, we will include a full citation to the dataset in the article if it is accepted for publication.

Reviewer 2 Report

Sections from lines 452 to 472 should be filled up.

While I was able to review the article's statistical modeling methodology, and I find it sound, formal and worthy of publication, I can not assess the specific relevance to the specific field of  the performance  evaluation of Libraries, Archives, and Museums websites (LAMs).

I recommend the final decision for the acceptance of the article to be subject to  the evaluation of the relevance of the publication to the specific application area.

Author Response

Reviewer 2

Reviewer 2 Comment 1:

Sections from lines 452 to 472 should be filled up.

Letter to the reviewers

First of all, we would like to express our appreciation to the reviewers for their valuable comments and observations. Our answers for all reviewers’ comments follow:

Response: We have updated the text of the paper.

Author Contributions: All authors have contributed equally. All authors have read and agreed to the published version of the manuscript.

Funding: The APC was funded by the Information Management Research Lab, Department of Archival, Library and Information Studies, University of West Attica.

Conflicts of Interest: The authors declare no conflict of interest.

Round 2

Reviewer 1 Report

Thank you very much. All of my previous comments were correctly addressed. Thank you very much for clarifying how the websites were cchecked. I think that the manuscript has been significantly improved. I wish you good luck in your future work.